# Regulation of Proteolytic Activity to Improve the Recovery of *Macrobrachium rosenbergii* Nodavirus Capsid Protein

**DOI:** 10.3390/ijms22168725

**Published:** 2021-08-13

**Authors:** Bethilda Anne Selvaraj, Abdul Razak Mariatulqabtiah, Kok Lian Ho, Chyan Leong Ng, Chean Yeah Yong, Wen Siang Tan

**Affiliations:** 1Department of Microbiology, Faculty of Biotechnology and Biomolecular Sciences, Universiti Putra Malaysia, Serdang 43400, Selangor, Malaysia; bethildaanne@yahoo.com (B.A.S.); yongcheanyeah@hotmail.com (C.Y.Y.); 2Department of Cell and Molecular Biology, Faculty of Biotechnology and Biomolecular Sciences, Universiti Putra Malaysia, Serdang 43400, Selangor, Malaysia; mariatulqabtiah@upm.edu.my; 3Laboratory of Vaccines and Biomolecules, Institute of Bioscience, Universiti Putra Malaysia, Serdang 43400, Selangor, Malaysia; 4Department of Pathology, Faculty of Medicine and Health Sciences, Universiti Putra Malaysia, Serdang 43400, Selangor, Malaysia; klho@upm.edu.my; 5Institute of Systems Biology (INBIOSIS), Universiti Kebangsaan Malaysia, Bangi 43600, Selangor, Malaysia; clng@ukm.edu.my; 6Centre for Virus and Vaccine Research, School of Medical and Life Sciences, Sunway University, Bandar Sunway 47500, Selangor, Malaysia

**Keywords:** nodavirus, capsid protein, protein expression, degradation, protease inhibitors

## Abstract

The causative agent of white tail disease (WTD) in the giant freshwater prawn is *Macrobrachium rosenbergii* nodavirus (MrNV). The recombinant capsid protein (CP) of MrNV was previously expressed in *Escherichia coli*, and it self-assembled into icosahedral virus-like particles (VLPs) with a diameter of approximately 30 nm. Extensive studies on the MrNV CP VLPs have attracted widespread attention in their potential applications as biological nano-containers for targeted drug delivery and antigen display scaffolds for vaccine developments. Despite their advantageous features, the recombinant MrNV CP VLPs produced in *E. coli* are seriously affected by protease degradations, which significantly affect the yield and stability of the VLPs. Therefore, the aim of this study is to enhance the stability of MrNV CP by modulating the protease degradation activity. Edman degradation amino acid sequencing revealed that the proteolytic cleavage occurred at arginine 26 of the MrNV CP. The potential proteases responsible for the degradation were predicted in silico using the Peptidecutter, Expasy. To circumvent proteolysis, specific protease inhibitors (PMSF, AEBSF and E-64) were tested to reduce the degradation rates. Modulation of proteolytic activity demonstrated that a cysteine protease was responsible for the MrNV CP degradation. The addition of E-64, a cysteine protease inhibitor, remarkably improved the yield of MrNV CP by 2.3-fold compared to the control. This innovative approach generates an economical method to improve the scalability of MrNV CP VLPs using individual protease inhibitors, enabling the protein to retain their structural integrity and stability for prominent downstream applications including drug delivery and vaccine development.

## 1. Introduction

*Macrobrachium rosenbergii* nodavirus (MrNV) causes the notorious white tail disease (WTD) in giant freshwater prawns (*Macrobrachium rosenbergii*). The disease was first reported in 1995 on the islands of Guadeloupe, French West Indies [1], and led to an alarming outbreak of MrNV at a global scale including Asian countries such as China, India, Thailand, and Malaysia [2,3,4,5]. The MrNV genome comprises two linear positive-sense single stranded RNAs; RNA 1 (3.2 kb) and RNA 2 (1.2 kb) which encode for the RNA-dependent RNA polymerase (RdRp) and capsid protein (CP), respectively [6].

The MrNV CP is composed of a single polypeptide containing 371 amino acid residues [6]. A recombinant MrNV CP harboring a 6xHis-tag and a myc epitope at its C-terminal end was successfully produced in *Escherichia coli* [6]. The recombinant protein with a molecular mass of approximately 46 kDa self-assembled into virus-like particles (VLPs) [6]. Recently, the MrNV CP has gained sizable attention due to its unique self-encapsulating feature of the highly basic N-terminus that folds inwards the VLP core structure, and readily binds to negatively charged molecules [7], hence, making it a potential candidate for nucleic acid and drug deliveries [8,9]. Additionally, the MrNV CP VLPs also serve as a promising platform for the development of anti-viral peptide [10] and multi-component vaccines [11,12,13], as well as the elucidation of the first three-dimensional structure of the native virus [14].

Intriguingly, besides the full-length MrNV CP band of 46 kDa, a smaller band of approximately 43 kDa was reported by Goh et al. [6] on SDS polyacrylamide gels stained with Coomassie Brilliant Blue (CBB). The latter is thought to be a proteolytic product of the N-terminal MrNV CP since it can be detected by both the rabbit anti-MrNV capsid protein serum and the anti-His monoclonal antibody [10,11,12,15]. Despite the impressive features of MrNV CP VLPs, the occurrence of proteolysis reduces significantly the recovery of the full-length MrNV CP, and subsequently affects the stability of the VLPs [6,9,10,11,12]. The degradation of the MrNV CP is postulated to be caused by the presence of host proteases during expression and purification, which can be inhibited with the addition of protease inhibitors. Protease inhibitors are molecules that bind specifically to protease enzyme active sites, resulting in the formation of stable inactive protease inhibitor complexes, while some inhibitors modify an amino acid residue at the catalytic site to halt the protease activity [16].

Although protease inhibitor cocktails are widely preferred for their ability to inhibit a broad range of proteases, the supplementation of these inhibitor cocktails is cost inefficient for frequent large-scale recombinant protein production, as compared to supplementation of individual protease inhibitors [17,18]. To emphasize, the protease inhibitor cocktail generally costs approximately three times more expensive than specific protease inhibitor such as E-64 at their respective functional concentration, according to the pricing and datasheet for protease inhibitor cocktail powder (P2714) and E-64 protease inhibitor (E3132) from Sigma-Aldrich (St. Louis, MO, USA). Therefore, we aimed to reduce the proteolytic degradation of MrNV CP and to identify the protease involved using specific protease inhibitors. Since proteases are often site-specific, identification of the proteolytic cleavage site is, therefore, crucial in finding the responsible protease and its corresponding protease inhibitor. As such, the specific proteolytic cleavage site was determined using the Edman degradation sequencing, and the protease(s) responsible to degrade the MrNV CP was predicted using a peptide characterization software; PeptideCutter, ExPASy [19]. Subsequently, the proteolytic activity was modulated by supplementing targeted protease inhibitors during pre- and post-purification process. The method established in this study can be used to enhance the yield of MrNV CP VLPs for drug delivery and vaccine development.

## 2. Results

### 2.1. Edman Degradation Sequencing

The purified MrNV CP was separated on a SDS-polyacrylamide gel, and staining of the gel with CBB revealed the expected 46 kDa band and a distinct degraded 43 kDa band (Figure 1a). The 43 kDa-protein band was subjected to the Edman degradation sequencing, and the amino acid sequence obtained was: Arg-Asn-Arg-Asn-Pro (Figure 1b), which corresponds to the 27–31 amino acid residues of MrNV CP (NCBI UniProt accession no. AHM92901). The calculated molecular mass of the cleaved 1–26 residues was indeed 3 kDa (http://pepcalc.com/, accessed on 23 January 2019 ), in agreement with the degradation of MrNV CP as observed in Figure 1a.

### 2.2. Identification of Proteases That May Be Responsible for the Degradation of Recombinant MrNV CP

The PeptideCutter program was employed to identify the proteases responsible for the degradation prior to residue-27 (Arg27) of MrNV CP as residues 1–26 were degraded. Three potential proteases were predicted for the degradation at amino acid residue-26 (Arg26); two serine proteases (arginine-C proteinase and trypsin) and one cysteine protease (clostripain) (Figure 2).

### 2.3. SDS-PAGE Densitometry Analysis with ImageJ

The MrNV CP calibration curve was established to quantitate the proteolytic degradation rate of MrNV CP. MrNV CP samples ranging from 0.5–3.0 µg/well were separated on a SDS-polyacrylamide gel [12% (*w*/*v*)]. Acquisition of an inverted image (Figure 3a) enabled an accurate integrated intensity quantification of the entire protein band in each sample using the ImageJ (Java-based image processing and analysis software) [20]. A calibration curve was plotted using the integrated intensity reading versus the amount of MrNV CP (Figure 3b). Linear regression data of the plot are summarized in Table 1. A good linear relationship was demonstrated with a correlation coefficient value of 0.9908.

After the linear calibration curve was obtained, MrNV CP samples were set to 1.5 µg/well for quantification. Figure 4 shows that the degradation rate profile of MrNV CP decreases from E-64 < AEBSF < PMSF < control. In the absence of protease inhibitors, only 34% of MrNV CP remained intact after 12 days. The addition of PMSF (serine protease inhibitor) did not significantly improve the amount of intact MrNV from days 3 to 12 as compared to the control. Supplementation of the samples with AEBSF (serine and cysteine protease inhibitor), significantly increased the amount of intact MrNV from days 6 to 12 as compared to the control. Interestingly, E-64 (cysteine protease inhibitor) showed the highest amount of intact MrNV CP recovered from 3 to 12 days post purification, indicating cysteine protease could be involved in degrading MrNV CP.

## 3. Discussion

The recombinant MrNV CP was produced using the *E. coli* expression system nearly a decade ago, but limited information is available on the stability of the protein. The purified MrNV CP degrades into a smaller protein band of 43 kDa. Apart from this degraded protein band, an additional protein band of 30 kDa was previously observed by Goh et al. [6] and Thong et al. [10] in their studies involving MrNV CP expressed in *E. coli* cells. Interestingly, this protein band is absent in MrNV CP produced in *Spodoptera frugiperda* (Sf9) cells [15] and native MrNV [10]. Therefore, we believe that this is an *E. coli* host protein which was packaged inside MrNV CP VLPs and co-purified as it does not exist in Sf9 cells and the native virus itself. Whilst the 43 kDa is a product of proteolytic degradation which affects the yield and stability of the VLPs. It is hypothesized that the host proteases are responsible for the degradation of MrNV CP. Hence, in order to investigate this hypothesis, this study was designed to determine the type of proteases that causes the degradation, and to supplement a suitable protease inhibitor to minimize proteolytic degradation of MrNV CP.

The full-length MrNV capsid polypeptide comprises two main domains: the shell (S) domain (amino acid residues 1–252) and protruding (P) domain (amino acid residues 253–371). The Edman degradation amino acid sequencing results of the 43 kDa protein band revealed that the proteolytic cleavage site is located between Arg26–Arg27, which falls at the N-terminal arm of the S domain. In particular, this region lies within the viral RNA binding site (residues 20–29 of MrNV CP), an Arg- and Lys-rich region, which is known to facilitate MrNV CP interactions with negatively charged RNA molecules during the viral morphogenesis [21]. In particular, the presence of this highly basic region can be alternatively used to interact with the negatively charged nucleic acids, which enables successful encapsulation of plasmid DNA into the VLPs [8]. In addition to nucleic acid encapsulation, Hanapi et al. [22] demonstrated that the 10-residue peptide, ^20^KRRKRSRRNR^29^, also plays a major role as a nuclear-targeting sequence which allows MrNV CP to enter the hosts’ nuclei. Truncation at this highly basic region would severely compromise the functions of MrNV CP, which limits its application in the development of nanoparticles for packaging negatively charged cargos, and interrupts its nuclear targeting ability. Proteolysis of recombinant proteins in an expression system is a common phenomenon. In general, *E. coli* is considered as a common host to produce recombinant proteins. According to Maurizi [23], approximately 3% of enzymes present in *E. coli* can be characterized as proteolytic enzymes. Moreover, overexpression of recombinant proteins may retard cell growth as a consequence of metabolic stress experienced by the host cell [24]. ATP-dependent Lon (La), trypsin-like serine protease (DegP), outer membrane protease (OmpT), and metalloprotease (Ci) are among the proteases present in *E. coli* [25,26,27]. In the current study, the PeptideCutter program predicted three potential proteases that are capable of cleaving the 26th residue of MrNV CP; arginine-C proteinase (a serine protease), trypsin (a serine protease) and clostripain (a cysteine protease). Besides its cost inefficiency, the supplementation of protease inhibitor cocktail contains a mixture of protease inhibitors, hence, it is not suitable to determine the specific protease responsible for the degradation. In comparison, the supplementation of specific protease inhibitors narrows down the possible protease. For these reasons, a serine protease inhibitor (PMSF) [28,29], a cysteine protease inhibitor (E-64) [16,30], and a cysteine and serine protease inhibitor (AEBSF) [18,31], were chosen and added individually during the cell lysis process and after the purification process.

Densitometry analysis of the proteolytic degradation rate revealed that at day 0, the freshly purified MrNV CP remained intact with slight degradation. Interestingly, a prolonged storage at 4 °C, the protein degraded continuously to a consistent smaller protein (43 kDa), from 81% of intact MrNV CP (46 kDa) on day 3 to merely 34% on day 12 in the absence of protease inhibitors. A similar trend of degradation was observed with MrNV CP supplemented with PMSF, exhibiting insignificant reduction of degradation. Its ineffectiveness to protect the capsid protein against proteolysis suggests that the cleavage at residue 26 of MrNV CP is unlikely caused by a serine protease. By contrast, the rate of MrNV CP degradation reduced significantly in the presence of E-64 and AEBSF, indicating that a cysteine protease is responsible for the proteolytic attack of MrNV CP at residue 26. In general, majority of proteases found in *E. coli* strains are serine proteases, as reported by a large body of literature [23,27,32]. This has led to the application of PMSF for the purification of MrNV CP in earlier studies [6,9,10,11,12,13,22]. However, the present study revealed that supplementation of the cysteine protease inhibitor, E-64, significantly improved the stability and yield of full-length MrNV CP. Ryan et al. [17] reported that the inhibitors added in an early stage of the protein purification, during cell lysis, may be lost during the process, resulting in proteolysis after purification. Hence, in the present study, re-application of inhibitors in the purified protein samples was able to protect MrNV CP against on-going proteolytic activity during storage. This approach significantly improved the yield of MrNV CP for downstream applications including developments of vaccines and nano-carriers for drug deliveries, as well as for structural analysis.

Based on the densitometry results, the intensity of two bands (46 kDa and 43 kDa) at day 12 is not equivalent to the intensity of one band (46 kDa) at day 0. At day 12, the combined intensity of both 46 kDa and 43 kDa bands was approximately half of that of the 46 kDa band at day 0. These results were observed in all treatment groups. Furthermore, despite a significant improvement of the yield of full-length MrNV CP upon supplementation of E-64, the on-going proteolytic activity after purification suggests that the proteases could be packaged inside MrNV CP VLPs, and co-purified with the particles (Scheme 1).This postulation is supported by the observation of a dodecahedral cage inside MrNV CP VLPs studied by cryo-electron microscopy [7,14]. In addition, the N-terminus of MrNV CP is located inside the VLPs [7,14], which can only be cleaved by proteases packaged inside the particles. Nevertheless, further studies are required to prove the postulation of the on-going proteolytic activity during the storage of MrNV CP.

We have previously performed deletion and point mutations of the first 29 residues of the MrNV CP [21]. Transmission electron microscopic analysis revealed that the truncated MrNV CP mutants, 29∆MrNV CP and 20–29∆MrNV CP, with 29 residues and 20–29 residues truncation, respectively, from the N-terminal region exhibited a tremendous reduction in VLP size of approximately 33% as compared to that of the full length MrNV CP. While substitution mutations of the sequence “R26R27R29” with alanine residues produced VLPs that were 11% smaller compared to that of the full-length MrNV CP. Additionally, 29∆MrNV CP and 20–29∆MrNV CP deletion mutants, respectively, contained only 11 ng and 10 ng RNA (in 1 µg MrNV CP) as compared to the full-length MrNV CP which encapsidated approximately 127.5 ng RNA. Replacements of the arginine residues at positions “R26R27R29” with alanine also reduced the amount of RNA packaged (90 ng RNA in 1 µg MrNV CP). These results demonstrated the importance of the highly positive charged segment of residues 20–29 in RNA binding and determination of the VLP size. In addition, Hanapi et al. [22] demonstrated that amino acids 20–29 are also responsible for the internalization of MrNV VLP into the host cell nucleus. Collectively, the abovementioned findings emphasize the significance of the 10-residue peptide sequence, ^20^KRRKRSRRNR^29^. Hence, since the protease cleavage site lies between Arg 26 and Arg 27, we predict that mutations of these residues would affect the function of the MrNV VLP. As an alternative to point mutations, our current study involves the supplementation of E-64 protease inhibitor that does not only enhance the yield and stability of the MrNV VLP but also preserves its functions for drug delivery and vaccine development.

## 4. Materials and Methods

### 4.1. Expression and Purification of MrNV CP

*E. coli* TOP 10 cells harboring the pTrcHis-TARNA2 were grown in Luria-Bertani (LB) broth (1 L) supplemented with ampicillin (50 µg/mL), and incubated for 2 h at 37 °C with shaking at 180 rpm. When the culture reached A_600_ around 0.6–0.8, the expression of recombinant protein was induced with isopropyl β-D-1-thiogalactopyranoside (IPTG, 1 mM) for 5 h at 25 °C with shaking at 220 rpm. Cells were harvested by centrifugation at 5400× *g,* for 10 min at 4 °C. The pelleted cells were re-suspended in lysis buffer (25 mM HEPES, 500 mM NaCl; pH 7.4) containing MgCl_2_ (4 mM), freshly prepared lysozyme (0.2 mg/mL), and DNase I (0.02 mg/mL) for 2 h at room temperature with continuous rotation. The cell suspension was sonicated at 200 Hz for 20 s for 10 cycles. After sonication, the cell suspension was centrifuged at 12,000× *g* for 10 min, and the supernatant was filtered using a 0.45 µm pore size filter. The crude lysate was then loaded onto a His-trap 1 mL-column (GE Healthcare, Buckinghamshire, United Kingdom). The weakly bound host proteins were removed with binding buffer A (25 mM HEPES, 500 mM NaCl, 50 mM imidazole; pH 7.4) and binding buffer B (25 mM HEPES, 500 mM NaCl, 200 mM imidazole; pH 7.4). The MrNV CP was eluted using elution buffer (25 mM HEPES, 500 mM NaCl, 500 mM imidazole; pH 7.4). The eluted sample fractions containing the MrNV CP were pooled and concentrated to 1 mL using a centrifugal protein concentrator with molecular mass cut-off 10 kDa (Merck Millipore, Burlington, MA, USA). The concentration of the purified protein was determined using the Bradford assay [33]. The molecular mass and purity of MrNV CP were analyzed with sodium dodecyl sulfate-polyacrylamide gel electrophoresis (SDS-PAGE) [12% (*w*/*v*)].

### 4.2. Identification of Proteolytic Cleavage Site by Edman Degradation Sequencing

The MrNV CP samples purified in the absence of protease inhibitors were stored for 12 days at 4 °C to allow the occurrence of proteolytic degradation. The degraded MrNV CP samples were separated using SDS-PAGE [12% (*w*/*v*)] at 16 mA for 90 min. Proteins on the gel were transferred to a polyvinylidene difluoride (PVDF) membrane using a semi-dry blotter (Trans-Blot SD, BioRad, California, USA) at 25 W for 35 min in the presence of blotting buffer (50 mM borate/L, 10% methanol (*v*/*v*); pH 9.0). The membrane was stained with staining solution [0.1% (*w*/*v*) Coomassie Brilliant Blue R-250, 40% (*v*/*v*) methanol, 10% (*v*/*v*) acetic acid] for 3 min and de-stained with a de-staining solution [40% (*v*/*v*) methanol, 10% (*v*/*v*) acetic acid]. The membrane was air-dried and the smaller protein band (43 kDa) was excised from the membrane. Then, the excised band was subjected to amino acid sequencing using the Procise 492 Edman Micro Sequencer (Applied Biosystems, Foster city, CA, USA), which utilizes phenylisothiocyanate (PITC) reagent to interact with the last N-terminal amino acid residue of the 43 kDa MrNV CP, forming a PITC-amino acid complex. Under acidic conditions, the PITC-amino acid residue was cleaved and converted to a stable phenylthiohydantoin (PTH)-amino acid residue. The PTH-amino acid derivative was identified using 140C PTH Amino Acid Analyzer (Applied Biosystems, Foster city, CA, USA). The PITC reaction was repeated for a total of five cycles for the identification of the N-terminal amino acid residues of the MrNV CP.

### 4.3. Identification of Proteases That May Be Responsible for the Degradation of MrNV CP

The potential proteases responsible for the proteolytic degradation of MrNV CP (accession no. AHM92901) were identified by analyzing the amino acid sequence of the protein with the PeptideCutter program (https://web.expasy.org/peptide_cutter/, accessed 23 January 2019).

### 4.4. Inhibitory Study of MrNV CP Using Protease Inhibitors

The protease inhibitors employed in this study were purchased from Merck (USA). The characteristics and concentrations of the protease inhibitors used to reduce the degradation of MrNV CP in this study are listed in Table 2. The inhibitors were applied individually to the *E. coli* cell suspension at the lysis step, along with MgCl_2_ (4 mM), freshly prepared lysozyme (0.2 mg/mL) and DNase I (0.02 mg/mL) and incubated for 2 h at room temperature with continuous rotation. The MrNV CP was purified and concentrated as described in Section 4.1. A second dose of protease inhibitors was supplemented to the purified protein samples, which were then stored at 4 °C. Degradation of full-length MrNV CP on days 0, 3, 6, 9 and 12, was determined using SDS-PAGE.

### 4.5. SDS-PAGE Densitometry Analysis by ImageJ

To generate the calibration curve, freshly purified MrNV CP samples were diluted to 0, 0.05, 0.10, 0.15, 0.20, 0.25, and 0.030 µg/µL, mixed with 6 × SDS loading buffer [375 mM Tris-HCl (pH 6.8), 6% (*w*/*v*) SDS, 4.8% (*v*/*v*) glycerol, 0.03% (*w*/*v*) bromophenol blue, 9% (*v*/*v*) β-mercaptoethanol] and boiled for 10 min. The diluted samples (10 µL) were separated with SDS-PAGE [12% (*w*/*v*)] at 16 mA for 90 min. Separated proteins on the gel were stained with staining solution [0.1% (*w*/*v*) Coomassie Brilliant Blue R-250, 40% (*v*/*v*) methanol, 10% (*v*/*v*) acetic acid] for 30 min and de-stained with de-staining solution [30% (*v*/*v*) methanol, 10% (*v*/*v*) acetic acid] until the protein bands became visible. The SDS-polyacrylamide gel was scanned with an Epson L3150 scanner (Epson, Nagano, Japan). The image acquisition by ImageJ software was conducted as described by Gallo-Oller et al. [36]. First, the gel was imaged at 300 dpi (dots per inch), followed by the conversion to an 8-bit format. The obtained image was then inverted to develop an image with clear detectable bands. Subsequently, the image was subjected to background subtraction through the rolling ball radius method at 50.0 pixels. The protein band of each sample was circumscribed with the rectangular ROI selection tool. Data were acquired as integrated intensity values.

Analysis of MrNV CP protein degradation rate was performed on days 0, 3, 6, 9, and 12. The amount of protein samples loaded into each well was standardized to 1.50 µg. The proteins were separated with SDS-PAGE [12% (*v*/*v*)] at 16 mA for 90 min. Each intact MrNV CP band (46 kDa) was individually circumscribed with the rectangular ROI selection tool as described above. The recovery of intact MrNV CP were calculated with Equations (1)–(3).
(1)Regression equation of calibration curve;y=Gradient of  calibration curve ×x
(2)$$Relative\;amount\;of\;intact\;MrNV\;CP\;\left(x,µg\right)=\frac{Intergrated\;intensity\;value\;of\;Intact\;MrNV\;CP\;protein\;band\;\left(y\right)}{\;\;\;Gradient\;of\;\;calibration\;curve\;\left(Equation.1\right)}$$
(3)$$Recovery\;of\;intact\;MrNV\;CP\;\left(\%\right)=\frac{Integrated\;intensity\;value\;of\;intact\;MrNV\;CP}{Amount\;of\;MrNV\;CP\;utilized\;in\;FWB\;\left(1.5\;µg\right)}\;\;\times 100\%$$

### 4.6. Statistical Analysis

Statistical analysis was performed using the Statistical Package for Social Sciences software version 21 (IBM Corporation, Armonk, NY, USA). Significant differences in the degradation rates of MrNV capsid protein at days 3, 6, 9, and 12 were determined using one-way analysis of variance via Duncan’s multiple range test, where *p* < 0.05 is considered statistically significant.

## 5. Conclusions

Despite the common production of MrNv CP VLPs in *E. coli* for a decade, the yield and stability of the VLPs have been compromised due to continuous proteolytic degradation. To our best knowledge, this current study is the first in over ten years to have successfully identify the presence of the 43 kDa band as the N-terminal degraded product. The specific cleavage site of protease on MrNV CP was identified with the Edman degradation sequencing. The proteases that degraded the MrNV CP at the specific cleavage site were predicted with the PeptideCutter program. In the present study, MrNV CP supplemented with E-64 (a cysteine protease inhibitor) during pre- and post-purification resulted in a significantly improved yield of full-length MrNV CP. Our findings revealed that MrNV CP treated with E-64 has maintained its protein integrity 12 days after purification. To further increase the stability of the MrNV CP, point mutations can also be performed to eliminate the specific protease degradation site. However, detailed characterization of the mutants is needed, as the mutations could also jeopardize the stability of the VLPs.

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
