# Peer review of "Regulation of Proteolytic Activity to Improve the Recovery of Macrobrachium rosenbergii Nodavirus Capsid Protein"

_ijms, 2021, doi:10.3390/ijms22168725_

Round 1

Reviewer 1 Report

The paper ‘Regulation of proteolytic activity to improve the recovery of  Macrobrachium rosenbergii nodavirus capsid protein’ provides a straightforward study to show that a 26 amino acid long N-terminal fragment is cleaved by a cysteine protease from E.coli. Given the current interest in RNA-based drugs and the potential use of the nodavirus capsid as a platform for new vaccines, the work presented here could be surprisingly significant.   

All the experimental procedures are clearly explained and are appropriately presented. The degradation studies seem to have been evaluated carefully with error bars that show the significance of the effect of the cysteine protease inhibitor.  It seems that only one concentration of the inhibitors were used for this experiment. It would be interesting to know if increasing the inhibitor concentration has any significant effect on the degradation rate.

The paper is clearly written though there are a few typos some of which are given below:

L20 have allured attracted widespread attention

L27  was  were predicted in silico

L31  2.3 folds

L39  the giant freshwater prawns

L71 for a frequent large

It would be useful to add the RNRNP sequence identified by the Edman degradation to a modified and annotated version of Fig 2.2

Author Response

Point-by-Point Response to Reviewer 1 Comments

Reviewer 1:

Comment 1: It seems that only one concentration of the inhibitors were used for this experiment. It would be interesting to know if increasing the inhibitor concentration has any significant effect on the degradation rate.

Response:

We kindly appreciate your comment. According to the product data sheet for E-64 provided, the effective concentration range is 1-10 µM (Sigma-Aldrich, n.d.-a), which is similar to that of Gold Bio (Gold Bio, n.d.). Furthermore, several studies by other researchers have maintained the highest recommended concentration of E-64 at 10 µM (Forman et al., 2009; Inui et al., 1997; Katunuma & Kominami, 1995; Kershaw et al., 2017). Hence, we have set 10 µM as the maximum concentration of E-64 supplemented during the pre- and post-purification process of MrNV CP.

References:

Forman, H.J.; Zhang, H.; Rinna, A. Glutathione: Overview of its protective roles, measurement, and biosynthesis. Mol Aspects Med. 2009, 30, 1–12. doi: 10.1016/j.mam.2008.08.006

Gold Bio. (n.d.). Gold Biotechnology Stock Solution 1 mM E-64 Stock Solution Instructions. Available from: https://www.goldbio.com/product/3808/e-64

Inui, T.; Ishibashi, O.; Inaoka, T.; Origane, Y.; Kumegawa, M.; Kokubo, T. Cathepsin K antisense oligodeoxynucleotide inhibits osteoclastic bone resorption. J Biol Chem. 1997, 272, 8109–12. doi: 10.1074/jbc.272.13.8109

Katunuma, N.; Kominami, E. Structure, properties, mechanisms, and assays of cysteine protease inhibitors: cystatins and E-64 derivatives. Methods Enzymol. 1995, 251, 382–97. doi: 10.1016/0076-6879(95)51142-3

Kershaw, C. M.; Evans, G.; Rodney, R.; Maxwell, W.M.C. Papain and its inhibitor E-64 reduce camelid semen viscosity without impairing sperm function and improve post-thaw motility rates. Reprod Fertil Dev. 2017, 29, 1107–14. doi: 10.1071/RD15261

Sigma-Aldrich. (n.d.-a) E-64 protease inhibitor | 66701-25-5 [Internet]. [cited 2021 Jun 10]. Available from: https://www.sigmaaldrich.com/MY/en/product/sigma/e3132?gclid=Cj0KCQjwzYGGBhCTARIsAHdMTQx48vJuwpz_Ub0ANHuY9x65MCBJSsE12BxVwv6LmpP4exycWdZkuhoaAkmUEALw_wcB

Comment 2: The paper is clearly written though there are a few typos some of which are given below:

L20 have allured attracted widespread attention

L27  was  were predicted in silico

L31  2.3 folds

L39  the giant freshwater prawns

L71 for a frequent large

Response: Thanks for pinpointing the typographical errors. We have rectified the errors accordingly.

  • "allured" has been replaced with "attracted" (page 1, Line 20 of the revised manuscript).
  • "was" has now been replaced with "were" (page 1, Line 27).
  • "folds" has been changed to "fold" (page 1, Line 32).
  • "the" has been deleted (page 1, line 39).
  • "a" has been removed (page 2, line 71).

Comment 3: It would be useful to add the RNRNP sequence identified by the Edman degradation to a modified and annotated version of Fig 2.2

Response: We kindly appreciate your suggestion. We have now added RNRNP sequence in the modified Figure 2 (page 5).

Reviewer 2 Report

Selvaraj and colleagues in their manuscript focused on the proteolytic maturation of M. rosenbergii nodavirus (MrNV) capsid protein (CP). While the topic is of high interest due to the potential application of MrNV-based VLP as a drug and RNA carrier, I have some major comments, which can be found below. Therefore, I recommend manuscript reconsideration after major revision.

MAJOR COMMENTS

  1. Page 2 Lines 70-72 – the authors stated that the current proteases inhibitors cocktails are cost inefficient for a frequent large scale recombinant protein production. Can authors estimate the financial benefits in case of single inhibitor (i.e., E-64) usage in comparison to the current inhibitor mixture?
  2. Figure 1A - there is some pale band below 30 kDa. Can authors identify this band? Can this band be used as an internal control of protein stability in your buffer for 12 days?
  3. I would encourage authors to include in their experiments positive controls, i.e. some recombinant protein with similar protease recognition sites and compare the proteolytic cleavage between the known protein and MrNV CP.
  4. Figure 4 – it would be much easier to follow the experiments, if proteolytic cleavage of MrNV CP was presented for each treatment at different time points as separate subfigures, i.e. the MrNV CP on day 0, 3, 6, 9 and 12 for control, then the same for PMSF, etc. In addition, it would be very helpful if authors showed the MrNV CP behaviour in the presence of protease inhibitors cocktails.
  5. Moreover, I would like to ask whether the intensity of two bands (46 kDa and 43 kDa) for control treatment at day 12 is equal to intensity of one band (46 kDa) at day 0? Is there any additional degradation of MrNV CP?
  6. Finally, to confirm the hypothesis of cysteine protease-mediated proteolytic cleavage of MrNV CP authors should mutate the 26th amino-acid residue and check whether this alteration affect the stability of MrNV VLPs.

Author Response

Point-by-Point Response to Reviewer 2 Comments

Reviewer 2:

Comment 1: Page 2 Lines 70-72 – the authors stated that the current proteases inhibitors cocktails are cost inefficient for a frequent large scale recombinant protein production. Can authors estimate the financial benefits in case of single inhibitor (i.e., E-64) usage in comparison to the current inhibitor mixture?

Response:

Many thanks for your comment. This comment is addressed with an estimated costing, supported with the necessary references listed below. The protease inhibitor cocktail powder (P2714), Sigma Aldrich (Millipore Sigma, Missouri, USA) costs USD 64.00 for 100 mL of working solution (Sigma-Aldrich, n.d.-b). Approximately 25 mL of the cocktail is required to inhibit protease activity in 4 g of cell lysate. Thus, it costs USD 16.00 per MrNV CP purification cycle (1 L culture). The specific E-64 protease inhibitor (E3132), Sigma Aldrich (Millipore Sigma, Missouri, USA)  costs USD 81.00 for 1 mg of lyophilized powder (Sigma-Aldrich, n.d.-a). The supplementation of 10 µM is sufficient for one protein purification cycle, hence, it only costs approximately USD 5.70 per purification cycle. This estimated cost signifies that the supplementation of specific protease inhibitor is cost efficient, saving up to 2.8 folds for each purification compared to inhibitor cocktail powder. A similar view was shared by Plaxton (2019) and Ritchie (2013) in their studies.

References:

Plaxton, W. C. Avoiding Proteolysis during the Extraction and Purification of Active Plant Enzymes. Plant and Cell Physiology, 2019, 60, 715–724. doi:10.1093/pcp/pcz028

Ritchie, C. Protease Inhibitors. Materials and Methods, 2013, 3, 169.  doi:10.13070/mm.en.3.169

Sigma-Aldrich. (n.d.-a). E-64 protease inhibitor | 66701-25-5. Retrieved June 10, 2021, from https://www.sigmaaldrich.com/MY/en/product/sigma/e3132?gclid=Cj0KCQjwzYGGBhCTARIsAHdMTQx48vJuwpz_Ub0ANHuY9x65MCBJSsE12BxVwv6LmpP4exycWdZkuhoaAkmUEALw_wcB

Sigma-Aldrich. (n.d.-b). Protease Inhibitor Cocktail powder for general use, lyophilized powder. Retrieved June 9, 2021, from https://www.sigmaaldrich.com/MY/en/product/sigma/p2714?context=product

Comment 2: Figure 1A - there is some pale band below 30 kDa. Can authors identify this band? Can this band be used as an internal control of protein stability in your buffer for 12 days?

Response: Many thanks for your comment. Based on our findings, the presence of the pale band with a Mr of approximately 30 kDa (Figure 1A) has been previously observed in studies involving  MrNV CP expressed in E.coli cells including Goh et al (2011, Figure 4) and Thong et al (2019, Figure 1). Interestingly, this protein band is absent in MrNV CP produced in Spodoptera frugiperda (Sf9) cells (Kueh et al., 2017, Figure 1) and native MrNV (Thong et al., 2019, Figure 1). Hence, we believe that this is an E. coli host protein which was packaged inside MrNV CP VLPs and co-purified as it does not exist in Sf9 cells and the native virus itself. As a foreign protein which presents in minute amount, the pale 30 kDa band is not suitable to be used as an internal control of protein stability.

Reference:

Goh, Z.H.; Tan, S.G.; Bhassu, S.; Tan, W.S. Virus-like particles of Macrobrachium rosenbergii nodavirus produced in bacteria. J. Virol. Methods. 2011, 175, 74–79. doi: 10.1016/j.jviromet.2011.04.021.

Kueh, C.L.; Yong, C.Y.; Masoomi, D.S.; Bhassu, S.; Tan, S.G.; Tan, W.S. Virus-like particle of Macrobrachium rosenbergii nodavirus produced in Spodoptera frugiperda (Sf9) cells is distinctive from that produced in Escherichia coli. Biotechnol. Prog. 2017, 33, 549–557. doi: 10.1002/btpr.2409.

Thong, Q.X.; Wong, C.L.; Ooi, M.K.; Kueh, C.L.; Ho, K.L.; Alitheen, N.B.; Tan, W.S. Peptide inhibitors of Macrobrachium rosenbergii nodavirus. J. Gen Virol. 2018, 99, 1227–1238. doi: 10.1099/jgv.0.001116

Comment 3: I would encourage authors to include in their experiments positive controls, i.e. some recombinant protein with similar protease recognition sites and compare the proteolytic cleavage between the known protein and MrNV CP.

Response: Thank you for your comment. We truly appreciate your suggestion. However, there are many parameters and complexities that must be considered carefully for this experiment. Firstly, the identification of a recombinant protein containing the exact cutting site as identified in MrNV CP is a major challenge. Next, if we consider cloning the specific protease cleavage site unto a known protein such as albumin, there are many intricate parameters to examine such as the differences in terms of stability of the cloned protein in a buffer system compared to its native counterpart. This factor must be taken into consideration to ensure accurate analysis of the degradation rate. Another crucial factor to consider is the high possibilities of the cloned protein to fold in different conformations, leading to spatial hindrances. Hence, this may cause inaccessibility for the protease present in MrNV CP to efficiently cleave the specific proteolytic site in the cloned recombinant protein.

Comment 4: Figure 4 – it would be much easier to follow the experiments, if proteolytic cleavage of MrNV CP was presented for each treatment at different time points as separate subfigures, i.e. the MrNV CP on day 0, 3, 6, 9 and 12 for control, then the same for PMSF, etc. In addition, it would be very helpful if authors showed the MrNV CP behaviour in the presence of protease inhibitors cocktails.

Response: We sincerely appreciate your comment. Nevertheless, the rationale of presenting different treatment groups at the same time point was for easier comparison of degradation rate.  In addition to that, presenting a particular treatment group at different time points would require different sets of the protein to be purified for a particular treatment group. For instance, to present the MrNV CP treated with PMSF on days 0, 3, 6, 9, and 12, on the same SDS-PAGE gel, five different sets of protein purification must be executed. This may lead to inaccuracy during data analysis as the degradation rate is quantified from multiple sets of purified protein instead of a single set, hence, preventing us to accurately study the correlation of the degradation rate at different time points. The main reason we excluded the supplementation of protease inhibitor cocktail as a treatment group is that it deviates from the scope of our study, which is to determine the specific protease responsible for the degradation of MrNV CP. Furthermore, incorporating the protease cocktail inhibitor is cost-inefficient for large scale purification of the MrNV CP as explained in comment 1, thus failing to serve a purpose for our study.

Comment 5: Moreover, I would like to ask whether the intensity of two bands (46 kDa and 43 kDa) for control treatment at day 12 is equal to intensity of one band (46 kDa) at day 0? Is there any additional degradation of MrNV CP?

Response:

Many thanks for your question. Our results showed that the intensity of two bands (46 kDa and 43 kDa) for the control treatment at day 12 is not equivalent to the intensity of one band (46 kDa) at day 0. At day 12, the combined intensity of both 46 kDa and 43 kDa bands was approximately half of that of the 46 kDa band at day 0. Additional random degradations were observed (Figure 4b, unedited and uncropped images) but too minute to be detected and quantified. Interestingly, these random degradations were not only observed in the control group, but also present in all treated samples in the form of faint bands from days 6 to 12. The occurrence of these on-going degradations could be due to the proteases packaged inside MrNV CP VLPs. The postulation of the on-going proteolytic activity is provided in the manuscript (page 7, Lines 198-206).

Comment 6: Finally, to confirm the hypothesis of cysteine protease-mediated proteolytic cleavage of MrNV CP authors should mutate the 26th amino-acid residue and check whether this alteration affect the stability of MrNV VLPs.

Response: Thank you for your kind suggestion. We are currently facing a lockdown imposed by the government to curb the spread of COVID-19. We are not allowed to enter our laboratory, and unsure when the lockdown will be lifted to enable us to provide this information. Kindly accept our sincere apologies. Although we are unable to proceed with the suggested experiment during this lockdown, we plan to study the role of Arg-26 and the highly basic region (20KRRKRSRRNR29) in VLP formation and stability in the future.

Round 2

Reviewer 2 Report

I would like to thank authors for their answers to my comments and suggestions. However, I regret that any of the author’s reply were not implemented in the final version of the manuscript. It is very probable that most of the readers will have the same questions as I did, and won’t find the answers in the publication.

In addition, I can understand that due to the COVID-19 pandemic it is hard to continue research, but it should not be the excuse of improper design of the experiments. In my opinion, i.e. in the article which describes the effect of single protease inhibitors on the MrNV CP stability, the comparison with a previously used protease inhibitor cocktail is a must

Moreover, in journals as the IJMS, it is necessary to confirm the findings with additional experiments, such as suggested mutagenesis. It is even more important since the authors declared that this experiment is planned by them.

I have an impression that the presented study is just a preliminary stage of the project and in my opinion, the current results are not enough for publication in the IJMS.

With regret, I cannot recommend the manuscript for publication in the International Journal of Molecular Sciences.

Author Response

Point-by-Point Response to Reviewer 2 Comments

Comment: I would like to thank authors for their answers to my comments and suggestions. However, I regret that any of the author’s reply were not implemented in the final version of the manuscript. It is very probable that most of the readers will have the same questions as I did and won’t find the answers in the publication.

Response: Many thanks for the comment. The estimated cost for protease inhibitor cocktail and E-64 protease inhibitor based on Sigma-Aldrich pricing has now been added to the revised manuscript (Page 2, Lines 72-76).

Comment: In addition, I can understand that due to the COVID-19 pandemic it is hard to continue research, but it should not be the excuse of improper design of the experiments. In my opinion, i.e. in the article which describes the effect of single protease inhibitors on the MrNV CP stability, the comparison with a previously used protease inhibitor cocktail is a must.

Response: Thank you for your suggestion for the extra experiment to compare the stability of MrNV CP using specific protease inhibitor and protease inhibitor cocktail. However, the proposed experiment using protease inhibitor cocktail is out of the scope of the present study. As stated in the manuscript, our objective is to determine the type of protease that causes the MrNV CP degradation at Arg26, through the protease inhibitors guided study. As protease inhibitor cocktail contains a mixture of different protease inhibitors, the use of the inhibitor cocktail will not be able to address our main research question, hence it was not included in the present study. For instance, protease inhibitor cocktail powder (P2714) from Sigma-Aldrich (Missouri, USA) contains AEBSF, Aprotinin, Bestatin, E-64, EDTA and Leupeptin (Sigma-Aldrich, n.d.), of which will complicate the whole study. We have also emphasized the reason of supplementing specific protease inhibitors instead of the protease inhibitor cocktail in the revised manuscript (Page 2, Lines 76-78) to avoid misinterpretations by readers.

Reference:

Sigma-Aldrich. (n.d.). Protease inhibitor cocktail powder for general use, lyophilized powder. Retrieved June 9, 2021, https://www.sigmaaldrich.com/MY/en/product/sigma/p2714?context=product

Comment: Moreover, in journals as the IJMS, it is necessary to confirm the findings with additional experiments, such as suggested mutagenesis. It is even more important since the authors declared that this experiment is planned by them.

Response: The current experiment was performed to identify the specific degradation site on MrNV CP using the Edman degradation amino acid sequencing method, and to determine the protease involved using specific protease inhibitors. The results of the present study pave the way for many experiments, which include several mutations and compare not only their degradation rate, but VLP assembly as well as the effects on structural and stability changes in order to confirm the usefulness of these new mutants for vaccine development and targeted drug delivery. Therefore, we have added a concluding remark suggesting these experiments as future perspective (Page 10, Lines 334-337).

Comment: I have an impression that the presented study is just a preliminary stage of the project and in my opinion, the current results are not enough for publication in the IJMS.

Response: With due respect, according to Reviewer 1, all the experimental procedures were clearly explained and are appropriately presented. The potential and significance of our study are highlighted by Reviewer 1 which reads "Given the current interest in RNA-based drugs and the potential use of the nodavirus capsid as a platform for new vaccines, the work presented here could be surprisingly significant”. Despite the common production of MrNv CP VLPs in E. coli for a decade, the yield and stability of the VLPs have been compromised due to continuous proteolytic degradation. To our best knowledge, this current study is the first in over ten years to have successfully identify the presence of the 43 kDa band as the N-terminal degraded product, which remained a mere hypothesis for many researchers in the past. We have also determined the proteolytic cleavage site via the Edman degradation amino acid sequencing, and have effectively pin-pointed the protease responsible for the degradation. The above-mentioned evidence proves that the manuscript describes a complete study. In addition, the findings reported in this manuscript pave the way for many future studies such as mutagenesis, VLP assembly and stability, structural changes, as well as their potential applications in vaccine development and drug delivery.  Also, the method established in this study can significantly reduce the production cost of future VLP-based therapeutics which face proteolytic stability challenges.

Round 3

Reviewer 2 Report

I would like to thank the authors once more for their answers to my comments. But still in my opinion, the manuscript does not fit to the International Journal of Molecular Sciences, with the current IF 5,923. An addition of few sentences in the discussion or conclusion section does not explain my concerns.

I strongly recommend to transfer the manuscript to other MDPI journals, i.e., Viruses or Pathogens.

Author Response

Once again, we thank reviewer 2’s comment. As the Editor also suggests us to add the contents of our rebuttal into the manuscript, we have revised our responses in Round 1 (15th June 2021) and added the information into our current revised manuscript. Responses in Round 2 were previously incorporated into the revised manuscript submitted on 27th June 2021.

Highlighted below are the responses that have been included in the revised manuscript.

Round 1 (15th June 2021)

Comment 2
Figure 1A - there is some pale band below 30 kDa. Can authors identify this band? Can this band be used as an internal control of protein stability in your buffer for 12 days?

Response: The purified MrNV CP degrades into a smaller protein band of ~43 kDa. Apart from this degraded protein band, an additional protein band of ~30 kDa was previously observed by Goh et. al [6] and Thong et. al [10]  in their studies involving MrNV CP expressed in E. coli cells. Interestingly, this protein band is absent in MrNV CP produced in Spodoptera frugiperda (Sf9) cells [15] and native MrNV [10]. Therefore, we believe that this is an E. coli host protein which was packaged inside MrNV CP VLPs and co-purified as it does not exist in Sf9 cells and the native virus itself. Whilst the ~43 kDa is a product of proteolytic degradation which affects the yield and stability of the VLPs. This information has now been added on page 6 (lines 149-157) of the revised manuscript.

Comment 4
Figure 4 – it would be much easier to follow the experiments, if proteolytic cleavage of MrNV CP was presented for each treatment at different time points as separate subfigures, i.e. the MrNV CP on day 0, 3, 6, 9 and 12 for control, then the same for PMSF, etc. In addition, it would be very helpful if authors showed the MrNV CP behaviour in the presence of protease inhibitors cocktails.

Response: In the current study, the PeptideCutter program predicted three potential proteases that are capable of cleaving the 26th residue of MrNV CP; arginine-C proteinase (a serine protease), trypsin (a serine protease) and clostripain (a cysteine protease). Besides its cost inefficiency, the supplementation of protease inhibitor cocktail contains a mixture of protease inhibitors, hence, it is not suitable to determine the protease responsible for the degradation. In comparison, the supplementation of specific protease inhibitors narrows down the possible protease. For these reasons, a serine protease inhibitor (PMSF) [28,29], a cysteine protease inhibitor (E-64) [16,30], and a cysteine and serine protease inhibitor (AEBSF) [18,31], were chosen and added individually during the cell lysis process and after the purification process. This current information has now been added on page 7 (lines 184-194) of the revised manuscript.

Comment 5  
Moreover, I would like to ask whether the intensity of two bands (46 kDa and 43 kDa) for control treatment at day 12 is equal to intensity of one band (46 kDa) at day 0? Is there any additional degradation of MrNV CP?

Response: Based on the densitometry results, the intensity of two bands (46 kDa and 43 kDa)  at day 12 is not equivalent to the intensity of one band (46 kDa) at day 0. At day 12, the combined intensity of both 46 kDa and 43 kDa bands was approximately half of that of the 46 kDa band at day 0. These results were observed in all treatment groups. This information has now been added on page 7 (lines 216-220) of the revised manuscript.

Round 2 (27th June 2021)

Comment 1
I would like to thank authors for their answers to my comments and suggestions. However, I regret that any of the author’s reply were not implemented in the final version of the manuscript. It is very probable that most of the readers will have the same questions as I did and won’t find the answers in the publication.

Response: To emphasize, the protease inhibitor cocktail generally costs approximately three times more expensive than specific protease inhibitor such as E-64 at their respective functional concentration, according to the pricing and datasheet for protease inhibitor cocktail powder (P2714) and E-64 protease inhibitor (E3132) from Sigma-Aldrich (Missouri, USA). These results were observed in all treatment groups. This information has now been added on page 2 (lines 72-76) of the revised manuscript.

Comment 2
In addition, I can understand that due to the COVID-19 pandemic it is hard to continue research, but it should not be the excuse of improper design of the experiments. In my opinion, i.e. in the article which describes the effect of single protease inhibitors on the MrNV CP stability, the comparison with a previously used protease inhibitor cocktail is a must.

Response: Therefore, we aimed to reduce the proteolytic degradation of MrNV CP and to identify the protease involved using specific protease inhibitors. This information has now been added on page 2 (lines 76-78) of the revised manuscript.

Comment 3
Moreover, in journals as the IJMS, it is necessary to confirm the findings with additional experiments, such as suggested mutagenesis. It is even more important since the authors declared that this experiment is planned by them.

Response: To further increase the stability of the MrNV CP, point mutations can also be performed to eliminate the specific protease degradation site. However, detailed characterization of the mutants is needed, as the mutations could also jeopardize the stability of the VLPs. This information has now been added on page 11 (lines 354-357) of the revised manuscript.

Round 4

Reviewer 2 Report

Thank you for the answers and comments.

Author Response

Thank you.